# Association of a Common *NOS1AP* Variant with Attenuation of QTc Prolongation in Men with Heroin Dependence Undergoing Methadone Treatment [note 1]

**DOI:** 10.3390/jpm12050835

**Published:** 2022-05-20

**Authors:** Kuan-Cheng Chang, Ke-Wei Chen, Chieh-Liang Huang, Wen-Ling Liao, Mei-Yao Wu, Yu-Kai Lin, Yi-Tzone Shiao, Wei-Hsin Chung, Yen-Nien Lin, Hsien-Yuan Lane

**Affiliations:** 1Division of Cardiovascular Medicine, Department of Medicine, China Medical University Hospital, Taichung 404332, Taiwan; kennychen0205@gmail.com (K.-W.C.); d12717@mail.cmuh.org.tw (Y.-K.L.); motic721@gmail.com (W.-H.C.); thinkingandwriting@gmail.com (Y.-N.L.); 2Graduate Institute of Biomedical Sciences, China Medical University, Taichung 404333, Taiwan; hylane@mail.cmuh.org.tw; 3Department of Addiction Treatment, Tsaotun Psychiatric Center, Ministry of Health and Welfare, Nan-Tou County 54249, Taiwan; psyche.hc@gmail.com; 4Center for Personalized Medicine, China Medical University Hospital, Taichung 404332, Taiwan; wl0129@mail.cmu.edu.tw; 5Graduate Institute of Integrated Medicine, China Medical University, Taichung 404333, Taiwan; 6School of Post-Baccalaureate Chinese Medicine, China Medical University, Taichung 40402, Taiwan; meiyaowu0919@gmaill.com; 7Department of Chinese Medicine, China Medical University Hospital, Taichung 404332, Taiwan; 8Center of Institutional Research and Development, Asia University, Taichung 413305, Taiwan; ytshiao@gmail.com; 9Department of Psychiatry, China Medical University Hospital, Taichung 404332, Taiwan

**Keywords:** methadone, *NOS1AP*, QTc interval, heroin-dependence

## Abstract

**Background:** The effects of methadone-induced severe prolongation of the corrected QT interval (QTc) and sudden cardiac death appear unpredictable and sex-dependent. Genetic polymorphisms in the nitric oxide synthase 1 adaptor protein (*NOS1AP*) have been implicated in QTc prolongation in general populations. We investigated whether common *NOS1AP* variants interact with methadone in relation to QTc prolongation in patients with heroin dependence. **Methods:** We genotyped 17 *NOS1AP* variants spanning the entire gene in heroin-dependent patients who received a 12-lead electrocardiography (ECG) examination both at baseline and during maintenance methadone treatment in Cohort 1 and only during maintenance methadone treatment in Cohort 2. The QT interval was measured automatically by the Marquette 12SL program, and was corrected for heart rate using Bazett’s formula. **Results:** Cohort 1 consisted of 122 patients (age: 37.65 ± 8.05 years, 84% male, methadone dosage: 42.54 ± 22.17 mg/day), and Cohort 2 comprised of 319 patients (age: 36.9 ± 7.86 years, 82% male, methadone dosage: 26.08 ± 15.84 mg/day), with complete genotyping data for analyses. Before methadone, the QTc intervals increased with increasing age (r = 0.3541, *p* < 0.001); the age-adjusted QTc showed dose-dependent prolongation in men (r = 0.6320, *p* < 0.001), but abbreviation in women (r = −0.5348, *p* = 0.018) in Cohort 1. The pooled genotype-specific analysis of the two cohorts revealed that the QTc interval was significantly shorter in male carriers of the rs164148 AA variant than in male carriers of the reference GG genotype (GG: n = 262, QTc = 423 ± 1.4 ms; AA: n = 10, QTc = 404.1 ± 7 ms, *p* = 0.009), according to univariate analysis. The QTc remained shorter in male carriers of the rs164148 AA variant compared to GG genotype (423 ± 1.4 ms vs. 405.9 ± 6.9 ms, *p* = 0.016) in multivariate analysis after adjusting for age and methadone dosage. A cut-off QTc interval of <410 ms identifies 100% of AA carriers compared to none of GG carriers when receiving a daily methadone dosage of 30.6 ± 19.3 mg. There was no significant gene-drug interaction in contributing to the adjusted QTc (*p* = 0.2164) in male carriers of the rs164148 variants. **Conclusions:** Carriers of a common *NOS1AP* rs164148 AA genotype variant were associated with a shorter QTc interval in men receiving maintenance methadone treatment. This genetic polymorphism attenuates the QTc-prolonging effect by methadone, and thus may explain at least in part the unpredictable and heterogeneous risks for severe QTc prolongation and sudden cardiac death in patients on methadone.

## 1. Introduction

Methadone, a synthetic μ-opioid receptor agonist that is widely used in the treatment of heroin dependence and chronic pain, has been associated with drug-induced prolongation of the heart rate-corrected QT interval (QTc) leading to torsades de pointes (TdP) ventricular tachycardia and sudden death in susceptible patients [1,2,3,4,5,6,7,8,9]. In patch-clamp experiments, methadone has been shown to block the *I*_Kr_ current encoded by the human *ether-a-go-go* related gene (*HERG*) at clinically relevant concentrations [10], thereby providing a plausible mechanism for QTc prolongation and malignant ventricular tachyarrhythmias observed in susceptible patients taking methadone.

Methadone is administered as a chiral mixture of *(R,S)*-methadone, and is extensively metabolized in the liver through the cytochrome P-450 (CYP) enzyme system. (*R)*-methadone, which exerts a clinical opioid effect, is preferentially metabolized by CYP2C19 [11]. Conversely, (*S)*-methadone, mainly metabolized by CYP2B6 owing to its stereoselectivity [11], blocks the *HERG* channel more potently and is primarily related to arrhythmic complications caused by drug-induced long QT syndrome. Although impaired metabolism of *(S)*-methadone in carriers with CYP2B6**6/***6* genotype may contribute to higher dose-normalized *(S)*-methadone plasma concentrations [12], and has been associated with greater QTc intervals in patients taking *(R,S)*-methadone [13], it has been shown that the CYP2B6**6/***6* genotype alone was not an independent predictor of QTc interval variations [14].

Apart from the genetic polymorphisms involving the methadone metabolic pathway, common variants in the nitric oxide synthase 1 adaptor protein gene (NOS1AP) have been implicated in QTc interval prolongation and risk of sudden cardiac death both in general populations and in patients with diabetes, coronary artery disease, and drug-induced long QT syndrome [15,16,17,18,19,20,21]. However, whether the genetic polymorphisms in *NOS1AP* can also modify the QTc intervals in patients undergoing methadone treatment remains largely unknown. In a translational animal model, we have previously shown that the mechanism through which NOS1AP acts on myocytes is to interact with NOS1 to regulate intracellular calcium and *I*_Kr_ current [22]. Since both methadone and *NOS1AP* can affect *I*_Kr_ current, we hypothesized that common *NOS1AP* variants may contribute to the heterogeneous changes in QTc intervals in patients taking methadone. We thus conducted a prospective cohort study to assess whether common *NOS1AP* variants interact with methadone in relation to QTc prolongation in patients with heroin dependence.

## 2. Materials and Methods

### 2.1. Study Subjects

We prospectively enrolled two cohorts of patients with heroin dependence who were followed-up in the outpatient methadone clinics at China Medical University Hospital in this study. These patients underwent genotyping for 17 common *NOS1AP* variants spanning the entire gene. In order to assess the effects of methadone on QTc interval, patients in Cohort 1 who received a 12-lead ECG examination both at baseline and during maintenance methadone treatment were enrolled. In order to assess the genetic interaction with methadone on QTc, we also enrolled patients who only underwent the ECG examination during maintenance methadone treatment as Cohort 2, in order to increase the sample size for a combined analysis. The Institutional Review Board for the Protection of Human Subjects of the China Medical University Hospital approved the study protocol (DMR94-IRB-007). Each subject signed the informed consent before inclusion. Heroin addiction was diagnosed by an experienced psychiatrist according to the Diagnostic and Statistical Manual of Mental Disorders, Fourth Edition (DSM-IV), published by the American Psychiatric Association in 1994. In order to be included, the study subjects had to meet the following criteria: duration of heroin addiction >1 year, aged ≥18 and ≤65 years, and good physical health as assessed by history taking and a comprehensive physical examination. Patients were excluded from the study if they currently had active physical or mental disorders, including significant cardiovascular disease, hepatic or renal dysfunction; mental retardation; or poly-substance abuse. Concomitant use of drugs or alcohol was assessed from patient reports. Medications that can potentially affect cardiac conduction, repolarization, or methadone metabolism were carefully screened to avoid confounding effects on the QTc intervals. All patients received oral methadone at a starting dose of 20 mg/day, followed by incremental increases in the daily dose. Doses were titrated to the maintenance levels based on consideration of the following factors, including self-reported heroin use, presence or absence of opioid withdrawal symptoms, and urine toxicology data. All study patients were followed-up regularly at the outpatient methadone clinics at an interval of 1 to 4 weeks while receiving methadone induction, titration, and maintenance treatment. We also performed telephone interviews for those patients who were lost to regular follow-up, in order to improve treatment adherence and to acquire information regarding any adverse effects experienced during methadone treatment.

### 2.2. Measurement of QTc Interval by 12-Lead ECG

All 12-lead ECGs were recorded by a GE Marquette’s MAC 5500 ECG recorder (GE Medical Systems, Milwaukee, WI, USA) using a standardized protocol with consistent and accurate localization of each electrode. A 10-s resting 12-lead ECG was recorded at a sampling frequency of 500 Hz, and was digitally transmitted and stored to the MUSE system (GE Marquette, MI, USA) at the core ECG laboratory of China Medical University Hospital for subsequent analyses. In Cohort 1, the baseline 12-lead ECG recording was performed before the observed intake of the participant’s first methadone dose. The ECG for analysis during maintenance methadone treatment was typically obtained 1 month after the initiation of the treatment protocol in Cohort 1 and Cohort 2 patients. All of the stored ECGs were carefully reviewed by a board-certified cardiologist blinded to the study. ECGs with technical errors and inadequate quality were excluded before being subjected to automatic measurements by the 2001 version of the Marquette 12SL program (GE Marquette). The machine-read QT interval measurements are based on a “Global Median” beat, a superimposition of 12 representative median beats, which has previously been described and verified extensively [17,23]. Briefly, the 12SL measures the earliest onset of the Q wave in any lead, and the latest offset of the T wave in any lead, in order to obtain the QT interval; it is then corrected for heart rate using Bazett’s formula (QTc = QT/RR^1/2^). The 12SL algorithm incorporates the U wave in the measurement of QT interval only if the U wave merges with the preceding T wave. Otherwise, the algorithm excludes the U wave from the QT measurement. ECGs that may affect QTc measurement were excluded from analyses if there were significant rhythm disorders, wide QRS duration ≥ 120 ms, and the presence of acute or recent myocardial infarction.

### 2.3. Genomic DNA Extraction, Genotyping, and Laboratory Tests

The genomic DNA was extracted from peripheral blood leukocytes using the Puregene kit (Gentra Systems, Minneapolis, MN, USA) in accordance with the manufacturer’s instructions. We genotyped 17 single-nucleotide polymorphisms (SNPs) spanning the entire *NOS1AP* gene, including the SNPs that have previously shown to be associated with the QTc interval in independent samples [15,16,18], or with those that were identified using GenBank along with a Japanese and Han Chinese SNP database. SNPs were genotyped using the GenomeLab SNPstream genotyping system (Beckman Coulter Inc, Fullerton, CA, USA) in 6 ng of genomic DNA as described previously [24]. The SNPstream web-based software was used to design the primers and tagged probes for multiplex polymerase chain reactions (PCRs) performed in a 384-well-plate (MJS BioLynx, Brockville, ON, Canada) (Appendix A). After PCR, 0.67 U Exonuclease I (Amersham Pharmacia, Buckinghamshire, UK) and 0.33 U shrimp alkaline phosphatase (Amersham Pharmacia) were added and incubated. Finally, tagged probes were extended with single TAMRA- or BODIPY-fluorescein-labeled nucleotide terminator reactions, and then spatially resolved by hybridization to the complementary oligonucleotides array (SNPware Tag array). The tag arrays were then imaged by a two-laser, two-color charged couple device-based imager (GenomeLab SNPstream array imager). The candidate *NOS1AP* SNPs were identified in each well by their position and fluorescent color according to the position of the tagged oligonucleotides. Genotypes were then generated according to the comparative fluorescent intensities of each SNP, and the geographical view processed by the computer software. All of the laboratory tests were performed on the first visit after obtaining the written informed consent from all study subjects in Cohort 1 and Cohort 2.

### 2.4. Statistical Analyses

Clinical characteristics for all study patients are presented as n (%), median and interquartile range (Q1–Q3), or mean and standard deviation (SD). Student’s *t*-tests or Mann–Whitney *U* tests were used as tests of significance while comparing continuous variables between Cohort 1 and Cohort 2. Genotype frequency distribution was tested for Hardy–Weinberg equilibrium with a chi-squared test. Differences between proportions were assessed by a chi-square test or by Fisher’s exact test. Scatter plots, Pearson’s correlation coefficients, and multiple linear regression analyses were performed in order to assess linear relationships between variables and QTc intervals. Covariates such as age, gender, and methadone dosage, were considered to be independent variables to adjust for the effects of SNPs on the QTc intervals in the multiple regression model. In order to examine the potential interactions between gene and methadone treatment on the adjusted QTc interval, genotypes and methadone doses were used as explanatory variables for computing the gene-dose interaction term by the PROC GLM model. For the 17 *NOS1AP* SNPs genotyped in the current study, both the total and male samples had a >90% power to detect a genetic effect of 4.4 ms QTc at a significance level of 0.05 [25]. All statistical analyses were conducted using the SAS software (version 9.4, SAS Institute Inc., Cary, NC, USA), and a *p*-value less than 0.05 (two-sided) was used as the level of significance.

## 3. Results

The demographics and clinical characteristics of the study patients are presented in Table 1. Cohort 1 comprises 122 heroin-dependent patients (103 men, 19 women, age 37.7 ± 8.1 years) undergoing a 12-lead ECG examination both at baseline and during maintenance methadone treatment. Cohort 2 consists of 319 patients (262 men, 57 women, age 36.9 ± 7.9 years) who only received a 12-lead ECG examination during maintenance methadone treatment. The mean age, male-to-female ratio, baseline heart rate, and QTc intervals were similar between the two cohorts of patients. The mean dosage of maintenance methadone treatment was higher in Cohort 1 than in Cohort 2 (42.5 ± 22.2 vs. 26.1 ± 15.8 mg/day, *p* < 0.001). The prevalence of hepatitis C infection was 80.7% and 80.9%, respectively, in Cohort 1 and Cohort 2, which was much higher than the estimated prevalence rate in the general population [26], and was associated with a high percentage of patients with abnormal serum levels of aspartate aminotransferase (46.1% vs. 49.2%), γ-glutamyltransferase (17.7% vs. 21.6%), and alanine aminotransferase (22.6% vs. 32%). The serum levels of blood urea nitrogen, creatinine, and electrolytes were essentially normal in all study subjects. The results of urine amphetamine tests were positive in 23.5% and 24.5% of patients, respectively, in Cohort 1 and Cohort 2. In all study patients, the most common use of concomitant medication was flunitrazepam (10.4%) followed by estazolam (0.7%). Among the concomitant medications used, only two patients in Cohort 2 prescribed with tramadol were identified to be associated with a possible risk of torsades de pointes, according to the Arizona Center for Education and Research on Therapeutics classification on QT prolongation risk (http://crediblemeds.org/healthcare-providers/drug-list?rf=All) (accessed on 7 February 2022).

In Cohort 1, the QTc intervals significantly prolonged compared to the baseline values after methadone treatment in total (427 ± 22.7 vs. 421 ± 23.2 ms, *p* = 0.005) and in male patients (426 ± 22.6 vs. 420 ± 22.3 ms, *p* = 0.003), but not in female patients (429 ± 23.7 vs. 428 ± 27.2 ms, *p* = 0.842) (Figure 1A). In order to assess the influences of age and methadone dosage on QTc intervals, a scatterplot was constructed. The distribution of the QTc intervals before or after the initiation of methadone treatment was plotted against the corresponding age or methadone doses, with a best-fitting line representing the age-specific or dose-response effect. A significant positive correlation between age and QTc interval was observed before the administration of methadone in Cohort 1 patients (r = 0.3541, *p* < 0.001) (Figure 1B). During maintenance methadone treatment, we found a significant positive correlation between methadone dose and the age-adjusted QTc interval in total (r = 0.4558, *p* < 0.001) and in males (r = 0.6320, *p* < 0.001), but a negative correlation in females (r = 0.5348, *p* = 0.018) (Figure 1C).

To explore the complex interactions among QTc interval, genetic polymorphisms in *NOS1AP*, age, gender, and methadone dosage, we combined Cohort 1 and Cohort 2 patients for this purpose. Table 2 shows the association between the *NOS1AP* gene variants and the QTc interval variations during maintenance methadone treatment after adjusting for age, gender, and methadone dose. Each of the 17 SNPs had a successful genotype call of >95%, and conformed to the Hardy–Weinberg equilibrium. There were no significant associations between these *NOS1AP* variants and QTc variations, except for the carriers of rs164148 AA genotype at the 3′ end showing a significantly shorter QTc interval than carriers of the rs164148 GG genotype. The rs164148 variant was in high linkage disequilibrium with eight other variants (rs1963645, rs1964052, rs164146, rs164147, rs1876986, rs164149, rs737641, rs164151) covering a 4–kb region at the 3′ end of the *NOS1AP* with a pairwise r^2^ value of >87% by Haploview analyses (http://www.broad.mit.edu/mpg/haploview/) (accessed on 7 February 2022).

In univariate analysis, a significant association at α = 0.05 between carriers of rs164148 AA genotype and shorter QTc interval compared with the QTc in carriers of the rs164148 GG genotype was observed in total (406.8 ± 6.7 vs. 423.8 ± 1.3 ms, *p* = 0.013) and in male patients (404.1 ± 7 vs. 423.8 ± 1.3 ms, *p* = 0.009), but not in female patients (434 ± 22.2 vs. 427.9 ± 3 ms, *p* = 0.787). The genotype-specific association of shorter QTc interval in carriers of the rs164148 AA variant than in carriers of rs164148 GG genotype remains significant at α = 0.05 in multivariate analyses after adjusting for age, gender, and methadone dose, both in total (410.2 ± 6.8 vs. 425.9 ± 1.6 ms, *p* = 0.022) and in male participants (405.9 ± 6.9 vs. 423 ± 1.4 ms, *p* = 0.016) (Table 3). However, none of these associations was significant by using a Bonferroni-adjusted *p* value of 0.003 (0.05/17). Figure 2A depicts a cut-off adjusted QTc interval of ≤410 ms in identifying 100% of rs164148 AA carriers compared to none of the rs164148 GG carriers when receiving methadone at 30.6 ± 19.3 mg/day in male patients.

To further explore the interaction between the rs164148 *NOS1AP* variant and methadone dose in relation to the adjusted QTc intervals during maintenance methadone treatment in male participants, the corresponding QTc interval from each genotype carriers taking a projected methadone dose of 40 mg/day, 60 mg/day, and 80 mg/day, was compared in the PROC GLM model in Figure 2B. There was no significant gene-drug interaction in contributing to the adjusted QTc intervals (P gene*methadone = 0.216) in carriers of the rs164148 AA genotype when compared to that observed for non-carriers (Figure 2B).

## 4. Discussion

To the best of our knowledge, this is the first report that a common *NOS1AP* variant attenuates QTc prolongation in men with heroin dependence undergoing methadone treatment. The principal findings of the current study include the following: (1) methadone treatment at a relatively lower dose of 42.5 ± 22.2 mg/day was associated with a significant QTc prolongation compared to the baseline value, particularly in men; (2) there was a significant dose-dependent prolongation effect of methadone in men, but not in women; (3) a common *NOS1AP* variant rs164148 at 3′ end was associated with attenuation of QTc prolongation by methadone in men, which might partly explain the heterogeneous effects of QTc prolongation observed after methadone treatment; (4) a cut-off adjusted QTc interval of ≤410 ms could identify 100% of rs164148 AA carriers compared to non-carriers; (5) although the current study shows no significant drug-gene interaction between methadone and the rs164148 G>A variant, this does not preclude future identification of other genetic variants in *NOS1AP* that may interact with methadone in modifying QTc intervals, particularly in different ethnic populations.

### 4.1. Methadone-Associated QTc Prolongation and Sudden Cardiac Death

Methadone, an opioid analog with an elimination half-life of 24–36 h, is a *HERG* channel blocker [10] that has been related to QT prolongation in a dose-dependent manner. We have previously shown a significant dose-dependent interaction between methadone and QTc in individuals receiving a lower median methadone dose of 40 mg/day (interquartile range: 30–60 mg/day) [27]. For patients receiving a higher daily methadone dose of 100 mg or greater, most previous studies also reported a positive correlation between methadone dose and QTc interval [4,5,6,9,28]. All of these findings indicate a significant association between methadone and QTc interval across a wide therapeutic range of methadone doses among different ethnic populations. Clinically, a QTc interval of ≥500 ms is indicative of a high risk for TdP ventricular tachycardia and sudden death [29,30], which was observed in 2–16% of patients who received methadone at median doses of ≥100 mg/day [4,5,6]. Although a daily methadone dose of <120 mg is rarely associated with severe QTc prolongation (>500 ms) [2,6,31], Ehret et al. [4] studied 167 hospitalized patients receiving methadone treatment for intravenous drug addiction, and found that the lowest daily dose associated with a QTc of 500 ms or longer was 30 mg. Documented TdP ventricular tachycardia has been observed in patients undergoing methadone treatment with doses ranging from 40 mg/day to 200 mg/day [4]. Furthermore, taking methadone may not have universal effects in prolonging QTc intervals when compared to their baseline values. We previously showed that after initiation of methadone therapy (median dose, 40 mg/day), the changes in QTc intervals are heterogeneous, with 61% of patients exhibiting an increase in QTc interval and 36% showing a decrease or no change (3%), when compared to the baseline values in patients with heroin dependence [27]. In a prospective study, Cruciani et al. [28] reported that the percentage of patients with QTc lengthening after methadone treatment was only 33% in a population receiving a higher dose of methadone (median dose, 110 mg/day) for opioid dependence or chronic pain. All of these findings suggest that genetic backgrounds may play an important role in the modification of QTc changes and the association of ventricular tachyarrhythmias or sudden cardiac death by methadone.

### 4.2. NOS1AP, a Genetic Modifier in Methadone-Associated QTc Prolongation

Arking et al. [15] first reported the association of a common variant of *NOS1AP* (rs10494366) in the non-coding region with QTc interval variation in a community-based German population. This genetic implication not only has been replicated in other racial/ethnic populations [18,32,33,34], but also been extended to a variety of clinical settings, including diabetes mellitus, coronary artery disease, myocardial infarction, and congenital or drug-induced long QT syndrome [17,18,19,21,35,36]. In a translational animal model, we have shown that *NOS1AP* is expressed in the heart, and interacts with NOS1-NO pathways to modulate cardiac repolarization via suppressing the sarcolemmal L-type calcium current and enhancing the *I*_Kr_ current [22]. Because both methadone and *NOS1AP* have direct effects on the *I*_Kr_ current, it is possible that the heterogeneous responses of QTc changes observed after methadone treatment may be attributable, at least in part, to the interplays between methadone and *NOS1AP*.

In the current study, we found that carriers of rs164148 AA genotype at the 3′ end in *NOS1AP* were associated with shorter QTc intervals compared with carriers of the rs164148 GG genotype in men undergoing methadone treatment. In these patients, a cut-off QTc interval of ≤410 ms sufficiently discriminates the two genotypes, indicating that a minority of patients carrying the AA genotype (2.5% frequency) might be protected from clinically extreme methadone-associated QTc prolongation. Our finding also concurs with the previous report that the preponderance of significant SNP-QT associations in Caucasian and Hispanics populations were at the 5′ end of *NOS1AP,* in contrast to the significant associations in Chinese population at the 3′ end [37]. Thus, a heterogeneous influence of common *NOS1AP* variants on QT interval may occur across races and ethnicities [37].

In the current study, there was no significant rs164148 SNP-QTc association in female patients, which was apparently underpowered because of the limited samples of female patients with heroin dependence. Nevertheless, a strong association of the *NOS1AP* SNP, rs10494366, on QTc interval in women but not in men has been reported in two independent populations [38]. However, previous studies of the gender-specific effects of *NOS1AP* on QTc have been inconclusive [32,39,40]. In a genome-wide association study, Arking et al. reported a stronger *NOS1AP*-QTc association in women than in men in American adults of European ancestry, but not in a Southern German population [15]. Post et al. found no significant gene-gender interaction in a QTc association study of four candidate SNPs in old order Amish families [39]. Indeed, further studies are needed to investigate whether the gender differences in *NOS1AP*-QTc associations are evident across diverse populations with or without exposure to QT-prolonging drugs such as methadone. Since there was no significant drug-gene interaction in male patients, independent influence of methadone (QTc prolongation) and rs164148 SNP (QTc abbreviation) likely contributes to the final QTc interval in these patients. Future large-scale genome-wide association studies are required to identify other genetic loci in *NOS1AP* that may interact with methadone in regulating the QTc intervals in different ethnic populations.

### 4.3. Clinical Implications

Patients on methadone treatment are at increased risks for developing serious QTc prolongation and sudden cardiac death; however, the susceptibility is often unpredictable. This study first demonstrates that a genetic polymorphism in *NOS1AP* may attenuate methadone-associated QTc prolongation, which might confer a protective effect to some extent, of drug-induced long QT syndrome and sudden cardiac death in men with heroin dependence taking methadone. This result may also have important pharmacogenomic implications for predicting individual risk of developing serious QT prolongation and/or sudden arrhythmic death when taking different kinds of QT-prolonging drugs.

### 4.4. Limitations

Our study has several limitations. First, we did not measure serum levels of methadone to study the interaction between methadone concentration and the *NOS1AP* gene on cardiac repolarization manifested by the QTc intervals while they were receiving stable maintenance treatment. However, it has been reported that both oral methadone dose and the serum level of methadone correlate well with the variations in QTc intervals [14]. Second, although great efforts have been made to minimize the confounding effects on QTc intervals, it is possible that there are still other factors not identified during the study that may have affected the QTc statistics. Third, potential day-to-day variations in QTc intervals should be considered as only a single 12-lead ECG was used in our study for QTc measurement at the baseline and during maintenance methadone treatment. Fourth, there is a remarkably low number of women enrolled in the study, which is an inherent limitation to drug-addiction studies; thus, the results derived from female subjects may be underpowered. Finally, although we have identified a nominal association of common *NOS1AP* variant rs164148 with QTc abbreviation in patients receiving methadone treatment, the association was not significant after Bonferroni correction. It is possible that inadequate α error protection might lead to false positive results in the current study. However, similarly with our findings, Shah et al. also reported an SNP-trait association between *NOS1AP* variants at the 3′ end and QTc interval with nominally significant association, but not after Bonferroni correction in the Chinese population. Furthermore, multiple comparison adjustments, including Bonferroni adjustment, have been considered as too conservative in SNP-trait association study, particularly when the analyzed SNPs were in linkage disequilibrium [37]. Given that the genetic effects of *NOS1AP* on QTc differ across ethnic populations, and that the maintenance methadone dose also varies from study to study, results of the current study should be confirmed in further large-scale studies.

## 5. Conclusions

Carriers of a common *NOS1AP* rs164148 AA genotype variant were associated with a shorter QTc interval in men undergoing maintenance methadone treatment. This genetic polymorphism attenuates the QTc-prolonging effects by methadone, and thus may explain at least in part the unpredictable and heterogeneous risks for severe QTc prolongation and sudden cardiac death in patients on methadone. Further studies are necessary to explore the underlying molecular mechanisms responsible for the dual influences from methadone and *NOS1AP* on cardiac repolarization.

## Figures and Tables

**Figure 1 jpm-12-00835-f001:**
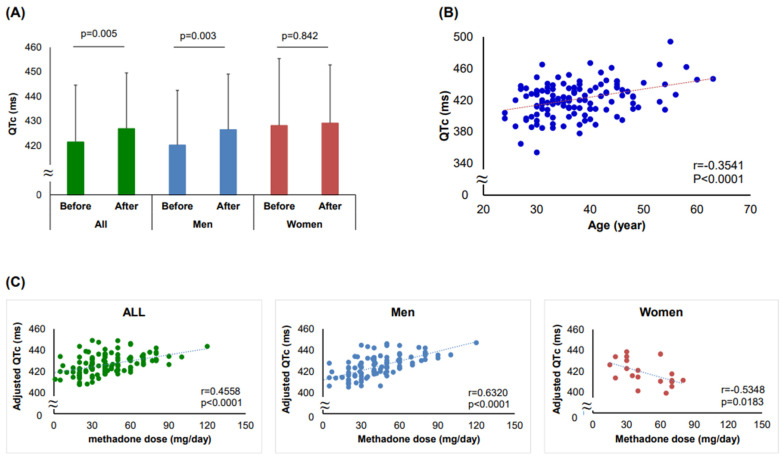
**Effects of methadone maintenance treatment, methadone dose, and age on QTc intervals.** (**A**) In Cohort 1, the QTc intervals significantly prolonged compared to the baseline values after methadone treatment in total (427 ± 22.7 vs. 421 ± 23.2 ms, *p* = 0.005) and in male patients (426 ± 22.6 vs. 420 ± 22.3 ms, *p* = 0.003), but not in female patients (429 ± 23.7 vs. 428 ± 27.2 ms, *p* = 0.842). (**B**) A significant positive correlation between age and QTc interval was observed before the administration of methadone in Cohort 1 patients. (**C**) During maintenance methadone treatment, we found a significant positive correlation between methadone dose and the age-adjusted QTc interval was observed in total and in male patients, but a negative correlation in females.

**Figure 2 jpm-12-00835-f002:**
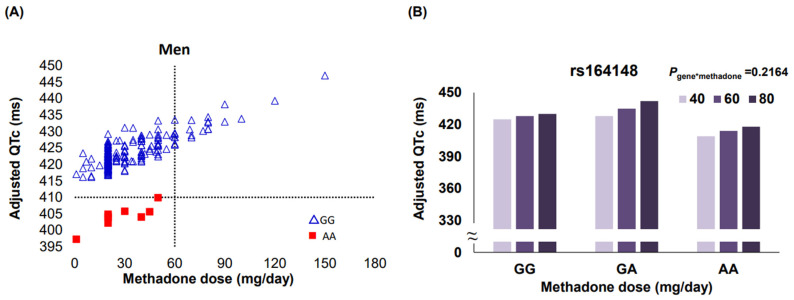
Association of *NOS1AP* rs164148 AA genotype with QTc intervals and analysis of *NOS1AP*-methadone interaction in men on methadone. (**A**) A cut-off adjusted QTc interval of ≤410 ms identified 100% of rs164148 AA carriers compared to none of the rs164148 GG carriers when receiving methadone at 30.6 ± 19.3 mg/day in male patients. (**B**) The PROC GLM model was used to explore the interaction between the rs164148 NOS1AP variant and methadone dose in relation to the adjusted QTc intervals during maintenance methadone treatment in male participants, the corresponding QTc interval from each genotype carriers taking a projected methadone dose of 40 mg/day, 60 mg/day, and 80 mg/day. There was no significant gene-drug interaction in contributing to the adjusted QTc intervals (P gene*methadone = 0.216) in carriers of the rs164148 AA genotype when compared to that observed for non-carriers.

**Table 1 jpm-12-00835-t001:** Demographics and clinical data of study patients.

Variables	Cohort 1 (*n* = 122)	Cohort 2 (*n* = 319)	Total (*n* = 441)	*p*-Value
**Age (years)**	37.65 ± 8.05	36.9 ± 7.86	37.1 ± 7.91	0.373
**Sex**				
Male	103 (84.43%)	262 (82.13%)	365 (82.77%)	0.568
Female	19 (15.57%)	57 (17.87%)	76 (17.23%)	
**Methadone dose (mg/day)**	42.54 ± 22.17	26.08 ± 15.84	30.64 ± 19.26	** *<0.001* **
**QTc (ms)**	426.85 ± 22.73	422.7 ± 22.2	423.85 ± 22.4	0.082
**QT (ms)**	380.42 ± 52.2	377.18 ± 28.03	378.08 ± 36.32	0.517
**HR (beats/min)**	75.14 ± 13.31	76.98 ± 13.1	76.48 ± 13.17	0.192
**BUN (mg/dL)**	11 (8–13)	10 (8–13)	10 (8–13)	0.206
**Creatinine (mg/dL)**	0.85 (0.74–0.95)	0.83 (0.73–0.93)	0.83 (0.73–0.93)	0.204
**Sodium (meq/L)**	138 (136–140)	138 (136–140)	138 (136–140)	0.545
**Potassium (meq/L)**	3.6 (3.4–4.2)	3.8 (3.5–4.1)	3.8 (3.5–4.1)	0.185
**Calcium (mg/dL)**	8.7 (8.5–9)	8.95 (8.6–9.15)	8.8 (8.6–9.1)	0.311
**ALT (U/L)**	30 (24–40)	31 (24–48)	31 (24–44)	0.637
**AST (U/L)**	33 (24–55)	34 (22–59)	33.5 (23–57)	0.596
**γGT (U/L)**	22 (16–36)	27 (17–47)	25 (17–44)	** *0.042* **
**ALT (range, 5–40 U/L)**				
Normal	89 (77.39%)	208 (67.97%)	297 (70.55%)	0.059
Abnormal	26 (22.61%)	98 (32.03%)	124 (29.45%)	
**AST (range, 5–34 U/L)**				
Normal	62 (53.91%)	156 (50.81%)	218 (51.66%)	0.571
Abnormal	53 (46.09%)	151 (49.19%)	204 (48.34%)	
**γGT (range, 8–50 IU/L)**				
Normal	93 (82.3%)	236 (78.41%)	329 (79.47%)	0.382
Abnormal	20 (17.7%)	65 (21.59%)	85 (20.53%)	
**HBs Ag**				
No	92 (80.7%)	254 (86.69%)	346 (85.01%)	0.129
Yes	22 (19.3%)	39 (13.31%)	61 (14.99%)	
**HCV Ab**				
No	22 (19.3%)	56 (19.11%)	78 (19.16%)	0.966
Yes	92 (80.7%)	237 (80.89%)	329 (80.84%)	
**Urine amphetamine test (>500 ng/mL)**				
Negative	88 (76.52%)	233 (75.16%)	321 (75.53%)	0.772
Positive	27 (23.48%)	77 (24.84%)	104 (24.47%)	
**Self-reported cocaine use**				
No	120 (100%)	318 (100%)	438 (100%)	1.000
Yes	0 (0%)	0(0%)	0 (0%)	
**Self-reported MDMA use**				
No	120 (100%)	318 (100%)	438 (100%)	1.000
Yes	0 (0%)	0(0%)	0 (0%)	
Self-reported ketamine use				
No	120 (100%)	318 (100%)	438 (100%)	1.000
Yes	0 (0%)	0 (0%)	0 (0%)	
**Taking concomitant medications**				
Estazolam	1 (0.82%)	2 (0.63%)	3 (0.68%)	
Flunitrazepam	17 (13.92%)	29 (9.12%)	46 (10.43%)	
Flunitrazepam/Estazolam	0 (0%)	1 (0.31%)	1 (0.23%)	
Flunitrazepam/Lorazepam	0 (0%)	1 (0.31%)	1 (0.23%)	
Flunitrazepam/Zolpidem	1 (0.82%)	0 (0%)	1 (0.23%)	
Tramadol	0 (0%)	2 (0.63%)	2 (0.45%)	
Zolpidem	0 (0%)	1 (0.31%)	1 (0.23%)	
Clonazepam	1 (0.82%)	0 (0%)	1 (0.23%)	
**Drugs with known TdP risk ^a^**	0 (0%)	0 (0%)	0 (0%)	1.000
**Drugs with possible TdP risk ^a^**	0 (0%)	2 (0.63%)	2 (0.45%)	
**Drugs with conditional TdP risk ^a^**	0 (0%)	0 (0%)	0 (0%)	1.000

All values are expressed as the mean ± SD, median (Q1–Q3), or n (%) as appropriate. ^a^: According to the classification of the Arizona Center for Education and Research on Therapeutics. Abbreviations: BUN, blood urea nitrogen; ALT, alanine aminotransferase; AST, aspartate aminotransferase; γGT, γ-glutamyltransferase; HBs Ag, Hepatitis B surface antigen; HCV Ab, hepatitis C antibody; MDMA, 3,4-methylenedioxy-N-methylamphetamine; TdP, torsades de pointes ventricular tachycardia.

**Table 2 jpm-12-00835-t002:** Association between *NOS1AP* variants and QTc intervals during maintenance methadone treatment after adjusting for age, gender, and methadone dose.

	Frequency_n (%)	HW_*p* Value	Adjusted QTc (ms)	*p*-Value
rs1415257				
GG	204 (46.7%)	0.663	425.5 ± 1.6	Ref
GA	192 (43.9%)		422.9 ± 1.6	0.247
AA	41 (9.4%)		419.5 ± 3.5	0.114
rs10494366				
GG	205 (46.7%)	1	425.3 ± 1.5	Ref
GT	190 (43.3%)		422.9 ± 1.6	0.279
TT	44 (10%)		420.1 ± 3.3	0.158
rs1572495				
CC	290 (67%)	0.655	423.9 ± 1.3	Ref
CT	127 (29.3%)		423.9 ± 2	0.991
TT	16 (3.7%)		421 ± 5.6	0.623
rs945713				
CC	246 (57.1%)	0.338	423.8 ± 1.4	Ref
CT	164 (38.1%)		423.6 ± 1.7	0.949
TT	21 (4.9%)		427.3 ± 4.9	0.491
rs1415263				
TT	134 (30.9%)		424.3 ± 1.9	Ref
TC	215 (49.7%)	0.888	424.4 ± 1.5	0.948
CC	84 (19.4%)		420.9 ± 2.4	0.270
rs6683968				
TT	133 (30.6%)		424.8 ± 1.9	Ref
TG	219 (50.3%)	0.671	424.6 ± 1.5	0.938
GG	83 (19.1%)		420.3 ± 2.4	0.158
rs2661818				
GG	291 (66%)		424.2 ± 1.5	Ref
GC	129 (29.3%)	0.177	423.5 ± 1.9	0.771
CC	21 (4.8%)		419.7 ± 2.4	0.371
rs3751284				
AA	133 (30.6%)		422 ± 2.3	Ref
AG	219 (50.3%)	0.264	424.5 ± 1.6	0.367
GG	83 (19.1%)		424.4 ± 1.9	0.414
rs1963645				
TT	288 (65.9%)		423.6 ± 1.3	Ref
TC	138 (31.6%)	0.244	425.7 ± 1.9	0.351
CC	11 (2.5%)		411 ± 6.7	0.065
rs1964052				
CC	317 (72.2%)		423.8 ± 1.2	Ref
CT	112 (25.5%)	1	424.4 ± 2.1	0.792
TT	10 (2.3%)		414 ± 7	0.171
rs164146				
GG	312 (71.4%)		423.8 ± 1.3	
GC	113 (25.9%)	0.647	425.2 ± 2.1	0.543
CC	12 (2.7%)		412.1 ± 6.4	0.076
rs164147				
CC	315 (72.2%)		423.8 ± 1.2	Ref
CA	111 (25.5%)	1	425 ± 2.1	0.623
AA	10 (2.3%)		415.7 ± 7	0.257
rs164148				
GG	316 (71.7%)		423.8 ± 1.2	Ref
GA	114 (25.9%)	0.842	425.5 ± 2.1	0.494
AA	11 (2.5%)		408.2 ± 6.7	** *0.022* **
rs1876986				
AA	111 (25.5%)		425.2 ± 2.1	Ref
AG	231 (53.1%)	0.182	424.2 ± 1.5	0.704
GG	93 (21.4%)		421.2 ± 2.3	0.201
rs164149				
AA	195 (44.6%)		424.8 ± 1.6	Ref
AG	203 (46.5%)	0.176	423.7 ± 1.5	0.608
GG	39 (8.9%)		421 ± 3.5	0.326
rs737641				
CC	92 (21.1%)		421.8 ± 2.3	Ref
CT	234 (53.7%)	0.116	424.2 ± 1.5	0.383
TT	110 (25.2%)		425.1 ± 2.1	0.290
rs164151				
TT	310 (71.8%)		423.5 ± 1.3	Ref
TC	111 (25.7%)	0.777	425.4 ± 2.1	0.449
CC	11 (2.5%)		412.2 ± 6.7	0.096

Abbreviations: HW, Hardy–Weinberg equilibrium; Ref, reference.

**Table 3 jpm-12-00835-t003:** Associations between *NOS1AP* rs164148 genotypes and QTc intervals by univariate analysis and multivariate analysis in total, in male, and in female patients.

SNP	Genotype	Frequency_n (%)	Univariate Analysis		Multivariate Analysis	
rs164148			Mean ± SE	*p* Value	Mean ± SE	*p*-Value
Total (N = 441)	GG	316 (71.7%)	423.8 ± 1.3		423.8 ± 1.2	
	GA	114 (25.9%)	425.5 ± 2.1	0.492	425.5 ± 2.1	0.494
	AA	11 (2.5%)	406.8 ± 6.7	** *0.013* **	408.2 ± 6.7	** *0.022* **
Male (*n* = 365)	GG	262 (71.8%)	423 ± 1.4		423 ± 1.4	
	GA	93 (25.5%)	425 ± 2.3	0.457	424.8 ± 2.3	0.491
	AA	10 (2.7%)	404.1 ± 7	** *0.009* **	405.9 ± 6.9	** *0.016* **
Female (*n* = 76)	GG	54 (71.1%)	427.9 ± 3		427.9 ± 3	
	GA	21 (27.6%)	427.8 ± 4.8	0.984	428.1 ± 4.9	0.967
	AA	1 (1.3%)	434 ± 22.2	0.787	431.1 ± 23.5	0.891

## Data Availability

The data presented in this study are available in the article and Appendix A.

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
