# Peer review of "Association of a Common NOS1AP Variant with Attenuation of QTc Prolongation in Men with Heroin Dependence Undergoing Methadone Treatment†"

_jpm, 2022, doi:10.3390/jpm12050835_

Round 1

Reviewer 1 Report

The work of Chang et al. presents the studies on the association of a common NOS1AP variant with a shorter QTc interval in humans undergoing maintenance methadone treatment. The obtained results can be very useful for clinicians and researchers. They are described correctly, arranged in a logical and coherent manner. The tables are legible, making it easier to follow the content of the work. I consider the manuscript to be well written and worthy of publication. I have only some minor suggestions:

  • Abstract: What does NOS1AP mean (line 3)? You should expand this abbreviation on first use. The same for ECQ (line 7).
  • Page 5: Enter the CYP abbreviation for cytochrome P-450.
  • Should not NOS1AP (gene) be written in italics like HERG?
  • Figures 1A and C: p (for probability) in capital letter. Change ALL to All, methadone to Methadone (in the left Figure 1C).
  • The name of γGT should be harmonized: γ-glutamyltransferase or gamma-glutamyltransferase.
  • There are some minor editing errors (e.g., punctuation) in the text that should be corrected before publication (follow the highlights in the manuscript).

Author Response

Response to Reviewers

Reviewer #1

The work of Chang et al. presents the studies on the association of a common NOS1AP variant with a shorter QTc interval in humans undergoing maintenance methadone treatment. The obtained results can be very useful for clinicians and researchers. They are described correctly, arranged in a logical and coherent manner. The tables are legible, making it easier to follow the content of the work. I consider the manuscript to be well written and worthy of publication. I have only some minor suggestions:

1. Abstract: What does NOS1AP mean (line 3)? You should expand this abbreviation on first use. The same for ECQ (line 7).

Response: We would like to thank the Reviewer for carefully reading our manuscript and making all those useful comments to help us improve it. We have used the full names of NOS1AP and ECG in “Abstract” (page 3, lines 4 and 9).

2. Page 5: Enter the CYP abbreviation for cytochrome P-450. 

Response: We thank the Reviewer for this comment. We have added the abbreviation for cytochrome P-450 in “Introduction” section (page 5, line 12).

3. Should not NOS1AP (gene) be written in italics like HERG?

Response: We appreciate the Reviewer’s constructive comments. We have rewritten NOS1AP in italics.

4. Figures 1A and C: p (for probability) in capital letter. Change ALL to All, methadone to Methadone (in the left Figure 1C).

Response: We thank the Reviewer for pointing out these errors. We have corrected “p” to “P” in Figure 1. We also changed “ALL” to “all” and “methadone” to “Methadone” in Figure 1.

5. The name of γGT should be harmonized: γ-glutamyltransferase or gamma-glutamyltransferase.

Response: We thank the Reviewer for this very kind comment. We have used γ-glutamyltransferase as the full name of γGT in the text (page 24, footnote of Table 1).

6. There are some minor editing errors (e.g., punctuation) in the text that should be corrected before publication (follow the highlights in the manuscript).

Response: We appreciate the Reviewer for pointing out the punctuation errors. We have corrected the errors in the “Title page” (page 1, line 4).

Reviewer 2 Report

The manuscript entitled: “Association of a Common NOS1AP Variant with Attenuation of QTc Prolongation in Men with Heroin Dependence Undergoing Methadone Treatment ” aimed to evaluate if genetic polymorphism in NOS1AP gene implicated in QTc prolongation in the general population can influence the appearance of methadone-induced severe QTc prolongation and sudden cardiac death in patients with heroin dependence. The manuscript is interesting and well written. Some aspects should be clarified before publication.

  1. Please clearly explain the rationale for choosing two cohorts of patients.
  2. As it can be seen in Table 1 the dose of methadone received by cohort 2 is significantly reduced compared with cohort 1. The main issue is if not the results of the correlation between the QTc is not influenced by these dose differences.
  3. In table 1 legend it should be clarified at which point were the tests done as cohort 2 does not have the results of the EKG from the baseline before methadone treatment.

Author Response

Response to Reviewers

Reviewer #2

The manuscript entitled: “Association of a Common NOS1AP Variant with Attenuation of QTc Prolongation in Men with Heroin Dependence Undergoing Methadone Treatment” aimed to evaluate if genetic polymorphism in NOS1AP gene implicated in QTc prolongation in the general population can influence the appearance of methadone-induced severe QTc prolongation and sudden cardiac death in patients with heroin dependence. The manuscript is interesting and well written. Some aspects should be clarified before publication.

1. Please clearly explain the rationale for choosing two cohorts of patients. 

Response: We thank the Reviewer for this important question, which we believe has helped to improve the clarity of our manuscript. We have provided a clear rationale for choosing two cohorts of patients in the “Methods” section to address the important question as follows: “To assess the effects of methadone on QTc prolongation, patients in Cohort 1 who received a 12-lead ECG examination both at baseline and during maintenance methadone treatment were enrolled. To assess the genetic interaction with methadone on QTc, we also enrolled patients who only underwent the ECG examination during maintenance methadone treatment as Cohort 2 to increase the sample size for a combined analysis.” (page 6, lines 17–22).

2. As it can be seen in Table 1 the dose of methadone received by cohort 2 is significantly reduced compared with cohort 1. The main issue is if not the results of the correlation between the QTc is not influenced by these dose differences.

Response: We totally agree with the Reviewer comment that the overall correlation between methadone dosage and the QTc intervals is not influenced by the dose differences between Cohort 1 and Cohort 2. We have addressed this issue in the “Discussion” section as follows: “We have previously shown a significant dose-dependent interaction between methadone and QTc in individuals receiving a lower median methadone dose of 40 mg/day (interquartile range: 30–60 mg/day) (Chang KC et al. JCE 2012). For patients receiving a higher daily methadone dose of 100 mg or greater, most previous studies also reported a positive correlation between methadone dose and QTc interval (Cruciani RA et al. J Pain Symptom Manage 2005; Ehret GB et al. Arch Intern Med 2006; Fanoe S et al. Heart 2007; Peles E et al. Addiction 2007; Anchersen K et al. Addiction 2009). All these findings indicate a significant association between methadone and QTc interval across a wide therapeutic range of methadone dose among different ethnic populations.” (page 13, line 25 to page 20, line 6).

3. In table 1 legend it should be clarified at which point were the tests done as cohort 2 does not have the results of the EKG from the baseline before methadone treatment.

Response: We thank the Reviewer for these valuable comments, which we think are very important to improve the clarity of the methodology and data presentation in this paper. We have added more detailed descriptions of ECG examination and the laboratory tests to clarify this issue in the “Methods” section. “In Cohort 1, the baseline 12-lead ECG recording was performed before the observed intake of the participant’s first methadone dose. The ECG for analysis during maintenance methadone treatment was typically obtained 1 month after the initiation of the treatment protocol in Cohort 1 and Cohort 2 patients.” (page 7, line 25 to page 8, line 4). Besides, under the new heading of “Genomic DNA extraction, genotyping, and laboratory tests” in the “Methods” section (page 8, line 18), we have added a sentence to describe the timing of laboratory examinations: “All of the laboratory tests were performed on the first visit after obtaining the written informed consent from all study subjects in Cohort 1 and Cohort 2.” (page 9, lines 14–16).

Reviewer 3 Report

Brief summary

The aim of the study was to investigate whether common NOS1AP variants interact with methadone in relation to QTc prolongation in patients with heroin dependence.

Findings

Carriers of a common NOS1AP rs164148 AA genotype variant was associated with a shorter QTc interval in men receiving maintenance methadone treatment. This genetic polymorphism attenuates the QTc-prolonging effect by methadone and thus may explain at least in part the unpredictable and heterogeneous risks for severe QTc prolongation and sudden cardiac death in patients on methadone.

Strengths

The topic is of interest and the results are of relevance. 

This study first demonstrates that a genetic polymorphism in NOS1AP may attenuate methadone-associated QTc prolongation, which might confer a protective effect, to some extent, of drug-induced long QT syndrome and sudden cardiac death in men with heroin dependence taking methadone.

Minor issues

There are a lot typographic errors. There are very long sentences. To improve readability, consider breaking them into multiple sentences. The authors are encouraged to proof-read thoroughly the text before resubmission. English must be excellent.

Author Response

Response to Reviewers

Reviewer #3

1. Minor issues

There are a lot typographic errors. There are very long sentences. To improve readability, consider breaking them into multiple sentences. The authors are encouraged to proof-read thoroughly the text before resubmission. English must be excellent.

Response: We are grateful to the Reviewer for this thoughtful comment. We have corrected the errors pointed out by the Reviewer. In addition, Grammar and language in the manuscript have been checked and edited by the professional Elsevier Language Editing service.

Round 2

Reviewer 2 Report

The authors addressed all my comments. The manuscript is ready for acceptance.